# Psychological Impact of the COVID-19 Pandemic on Pregnant Women: A Scoping Review

**DOI:** 10.3390/bs11120181

**Published:** 2021-12-16

**Authors:** Celia Campos-Garzón, Blanca Riquelme-Gallego, Alejandro de la Torre-Luque, Rafael A. Caparrós-González

**Affiliations:** 1Department of Nursing, Faculty of Health Sciences, University of Granada, 18071 Granada, Spain; celiacampos@correo.ugr.es (C.C.-G.); rcg477@ugr.es (R.A.C.-G.); 2Instituto de Investigación Biosanitaria ibs.GRANADA, 18012 Granada, Spain; 3Center of Biomedical Research in Mental Health, Department of Legal Medicine, Psychiatry and Pathology, Complutense University of Madrid, CIBERSAM, 28040 Madrid, Spain; af.delatorre@ucm.es

**Keywords:** pregnancy, coronavirus, mental health, depression, anxiety, stress

## Abstract

During the gestation period, pregnant women experience physical and psychological changes, which represent vulnerability factors that can boost the development of mental health conditions. The COVID-19 pandemic is producing new changes in the routines of the whole society, especially on lifestyle habits. The psychological impact associated with the COVID-19 pandemic and pregnant women remains unclear. A scoping review regarding the psychological impact of the COVID-19 pandemic on pregnant women was conducted. Searchers were conducted using the PubMed, Web of Science and CINAHL databases. Articles in Spanish, English and French were included. The search was conducted between November 2020 and September 2021. We identified 31 studies that evaluated 30,049 expectant mothers during the COVID-19 pandemic. Pregnant women showed high levels of anxiety and depression symptomatology. Fear of contagion and concerns regarding the health of the fetus were identified as the main variables related to psychological distress. An increase of the levels of depression, anxiety and stress during the COVID-19 pandemic amongst pregnant women has been observed. Moreover, an increased vulnerability of the fetus due to placental metabolic alterations is discussed. This review suggests that the COVID-19 pandemic is associated with a negative psychological impact on pregnant women. Thus, high levels of anxiety and depression symptoms suggest the need for a systematic approach.

## 1. Introduction

The first human cases of COVID-19 appeared back in 2019 [1]. As a result, the World Health Organization (WHO) declared a pandemic state in March 2020 [2,3,4]. Governments imposed a series of measures to reduce the risk of contagion, the spread of the SARS-CoV-2 virus and the increasing number of deaths. These measures included social distancing, quarantine and strict hygienic measures [5].

Countries around the world had to adapt to the rapid changes emerging in order to protect public health. Fear of contagion, the potential death of a relative and the possibility of being unemployed during this unprecedented times have affected mental health wellbeing [6]. 

In particular, psychological impairment was found to be more prevalent among vulnerable groups, including pregnant women [7]. During pregnancy, women may experience a series of physical and psychological changes, which directly affect their mental health [8]. Being multiparous, having a low level of education, being a teenager or having an unwanted pregnancy are among the factors that may affect pregnant women’s psychological state. Thus, a high risk pregnancy and having been diagnosed with a psychopathological disease before pregnancy are risk factors associated with a mental-health disorder during pregnancy [8,9]. In this respect, psychopathological symptoms such as anxiety and depression are the most frequently diagnosed. Moreover, adjustment disorders, substance abuse, eating and mood disorders can also appear during this period [10,11]. This symptomatology may affect both the fetus and the mother’s health: it boosts the risk of prematurity and low-birth weight, and increases the risk medical diseases during pregnancy such as gestational diabetes or pre-eclampsia [10,12,13,14]. Moreover, high levels of anxiety and depression during pregnancy have been associated with an excessive alcohol consumption [11,13]. 

Apart from the vulnerability factors associated with the expected changes of pregnancy [15], women may also experience increased psychological symptoms associated with the pandemic [16]. Uncertainty about the new virus, anxiety and fear have significantly affected the wellbeing of pregnant woman [17,18,19]. The main causes of anxiety, depression and stress were associated with fear of being infected while in public places, using public transport, during delivery at hospital, along with fear of vertical transmission of SARS-CoV-2 [20,21]. These variables have led women to avoid contacting hospitals/health units, withdrawing scheduled prenatal appointments, which ultimately led to reduced medical control during pregnancy [7,17]. Furthermore, fake information communicated through social media has increased pessimistic thoughts in pregnant women [7,22,23].

Additionally, certain social factors such as economic income and education level have also influenced coping strategies during the COVID-19 pandemic [24,25]. 

The aim of this study was to assess the psychological impact of the COVID-19 pandemic on pregnant women and to estimate the prevalence of mental symptoms in this population.

## 2. Materials and Methods

In order to perform an initial mapping of the literature, the methodology referring to a scoping review was used [26]. The characteristics of a scoping review are that it is systematic and rigorous. It allows the possibility of generating hypotheses, as well as proposing which areas of study are partially developed. This scoping review adheres to the PRISMA extension for scoping reviews [27]. The research question guiding this scoping review was to analyze the presence of mental symptoms in pregnant women associated with the COVID-19 pandemic.

### 2.1. Literature Search and Selection of Studies

Searches were conducted between November 2020 and September 2021 on PubMed, Web of Science and CINAHL databases. The search strategy included keywords related to psychological symptoms, pregnancy, postpartum, depression, trauma and coronavirus. MESH terms (e.g., “Pregnancy” [Mesh] AND “Coronavirus” [Mesh] AND (“Depression” [Mesh] OR “Depressive Disorder” [Mesh] OR “Anxiety” [Mesh] OR “Stress Disorders” OR “Traumatic” OR “Acute” [Mesh]) OR “Disorder” [Mesh]) and text word search terms (“pregnancy” AND “coronavirus” AND (“mental health” OR “depression” OR “anxiety” OR “stress”) were used. 

### 2.2. Inclusion and Exclusion Criteria

The inclusion and exclusion criteria is showed in Table 1.

### 2.3. Data Collection and Analysis

Eligible studies were selected through a multistep approach (elimination of duplicates, title reading, abstract, and full-text assessment). Two researchers (B.R-G.B. and C.C-G) independently examined titles and abstracts, evaluating afterwards full texts according to the inclusion criteria described above. Any disagreement between the reviewers was solved by means of a consensus session with a third reviewer (R.A.C-G). In case of ambiguity in reporting or lack of data, primary authors were contacted for clarification.

Searches yielded 87 unique articles. Forty-three were potentially eligible for inclusion based on title and abstract. After full-text review, thirty-one articles met the inclusion criteria. The PRISMA flow chart summarizes the study selection process (Figure 1).

### 2.4. Data Extraction and Management

Data were independently extracted by two researchers (B.R-G.B. and C.C-G), and the following information was considered for each article: (1) first author and year of publication; (2) study design; (3) the assessment instrument used in every study; (4) number of participants; (5) the average age of the study population; (6) the gestational age of pregnant women; (7) the percentage that was married when the study was conducted; (8) main results obtained and (9) the most important findings of each study. 

### 2.5. Quality Assessment Tool

The methodological quality of each study was assessed using the score in the Newcastle-Ottawa Quality Assessment Scale (NOQAS) [28] and the adapted version for cross-sectional studies. The criteria included 3 categories with a maximum score of 9 and 10 points for cohort and cross-sectional studies respectively. The first is the “selection”category, which accounts for a maximum of 4 points (5 points for cross-sectional studies), the second is the “comparability”category, which accounts for a maximum of 2 points, and the third is “outcome,” which accounts for a maximum of 3 points. Information regarding the quality of each study was included in Table 2.

## 3. Results

### 3.1. Study Characteristics

A total of 31 articles written in English were included in this review (Table 2). The countries where they were published were China (*n* = 5), Turkey (*n* = 4), United States (*n* = 2), Canada (*n* = 4), Iran (*n* = 1), Ethiopia (*n* = 1), Singapore (*n* = 1), Israel (*n* = 2), Spain (*n* = 3), Italy (*n* = 3), Japan (*n* = 1) Argentina (*n* = 1) and Qatar (*n* = 1). Twenty-six studies were cross sectional and three cohort study. A total of 30,049 women participated in the studies (*n* = 26,846 were pregnant women; *n* = 290 couples and *n* = 3,203 non-pregnant women who were controls). The mean age of the participants was 31.03 years old (SD = 4.93). The mean gestational age was 23.85 weeks of gestation (SD = 10.58). In addition, the psychological assessment of the participants was carried out through self-report measures during the COVID-19 pandemic.

### 3.2. Symptoms of Depression

Most of the selected studies reported depressive symptomatology among pregnant women related to the consequences of the COVID-19 pandemic [29,30,31,32,33,34,35,36,37,38,39,40,41,42,43,44,45]. The psychological tools included in these studies were: the Depression, Anxiety and Stress Scale-21 (DASS-21), the Edinburg Perinatal Depression Scale (EPDS), the Hospital Anxiety, Depression, Stress scale (HADS), The patient Health questionnaire (PHQ-9), Positive and Negative Affect Schedule (PANAS), Beck anxiety inventory (BAI), Inventory of Depression and Anxiety Symptoms II (IDAS-II), Mental Health Inventory (MHI-5), Depression, Anxiety and Stress Scale-21 (DASS-21), Edinburg depression scale (EDS), State-Trait Anxiety Inventory (STAI-State), Cambridge Worry Scale (CWS), Centre for Epidemiologic Studies Depression Scale (CES-D), Multidimensional Scale of Perceived Social Support (MSPSS), Impact of Event Scale-Revised (IES-R), Prenatal Distress Questionnaire (PDQ), Perceived Stress Scale (PSS), Connor-Davidson Resilience Scale (CD-RISC), Athens Insomnia Scale (AIS). Most of them were self-report questionnaires. The prevalence of depressive symptomatology was highly heterogeneous. Some studies reported a prevalence below 20% [40] while others reported a prevalence above 50% [46]. Maintaining a high partner satisfaction, having a high level of education, a high social support, staying physically active and a high income level appeared as variables that could decrease the levels of depression [29,38]. The use of informative tools provided by the hospital could also help to reduce the risk of depression [30,33,46]. When comparing the prevalence of depressive symptoms in pregnant and non-pregnant women, a variety of findings were reported. Thus, the prevalence was higher in non-pregnant women in two studies developed in Israel and China respectively [37,41]. Depressive symptoms were higher among pregnant women in a study in Argentina [45]. 

### 3.3. Symptoms of Anxiety

A total of 21 reports estimated the prevalence of anxiety in pregnant women during this pandemic [29,30,31,32,33,35,36,37,38,39,40,41,42,43,44,47,48,49,50,51,52,53,54,55,56]. Depression, Anxiety and Stress Scale-21 (DASS-21), the Edinburg Perinatal Depression Scale (EPDS), the Hospital Anxiety, Depression, Stress Scale (HADS), The Patient Health Questionnaire (PHQ-9), the General Anxiety Disorder-7 (GAD-7), the item 3 of the Oslo Social Support Scale (OSSS-3), the Self-Rating Anxiety Scale (SAS), the Trait Subscale of the Spielberg State-Trait anxiety Inventory (STAI-T), Beck Anxiety Inventory (BAI), the Inventory of Depression and Anxiety Symptoms II (IDAS-II), the Kessler Psychological Distress Scale (K10), the Mental Health Inventory—Short Form (MHI-5), the Patient-Reported Outcomes Measurement Information System (PROMIS), a modified version of the Pregnancy-related Anxiety Scale (PrAS), the Perceived Stress Scale (PSS), the Post-traumatic Stress Disorder Checklist 5 (PCL-5) and the Pandemic-related Pregnancy Stress Scale (PREPS) questionnaires were used to assess symptoms of anxiety, the majority self-report.

The scores obtained showed an increased mean score of anxiety in pregnant women as a result of the pandemic, (values; pre-COVID-19: 39.34 (SD = 6.39), COVID-19: 44.57 (SD = 9.55)) [54], (values; pre-COVID-19: -0.39 (SD = 0.04), COVID-19: 0,15 (SD = 0.03)) [51], values; pre-COVID-19: 184.78 (SD = 49.67), COVID-19: 202.57 (SD = 52.90) [47](values; pre-COVID-19: 20.6, COVID-19: 23.9) [39]. Most of these studies showed that more than one quarter of pregnant women were experiencing anxiety (17.2%; 18.1%; 32.2%; 32.7%; 35.8%) [29,30,40,52,57,58,59,60] and Sut et al., (2020) and Lebel et al., (2020) showed that more than half of the pregnant women presented this pathology (57%; 64.5% respectively) [38,46]. In contrast, one study comparing pregnant with non-pregnant women showed that the presence of anxiety symptoms was reduced during pregnancy (6.8% pregnant, 17.5% non-pregnant) [41]. The results from a prospective study reported that a great majority had believed that pregnant women have a higher risk for COVID-19 infection than general population. This cohort showed mean HADS-A score of 7.94 (SD = 4.03). Anxiety was associated with a high HADS-D score and concern about the inability to reach obstetrician, and being in advanced age [42]. A study carried out in 450 pregnant women and 274 after delivery reported a prevalence of anxiety symptoms above 50% [55]. Results from a study that evaluated anxiety from pre- to during-pandemic showed that 72% of women reported an indicative of moderate to high anxiety [40]. Another study reported a lower prevalence of clinically relevant anxiety levels in pregnant women in Italy (32.6%), but still significant [53]. Finally, a study from Spain showed higher levels of phobic anxiety in pregnant women than previous of the pandemic [35].

The main causes of anxiety reported in pregnant women during the pandemic were using a public transport (87.5%), COVID-19 infection of a family member (71.1%), being in public places (70%), concern for pregnancy complications and fetus health (70%), attending gynecological appointments (68.7%), becoming infected by COVID-19 (59.2%) and birth time (55.4%) [50]. The results reported from a cohort showed that most patients (82.5%) had concerns about infecting their babies during delivery [42].

### 3.4. Stress Levels

The assessment of stress in pregnant women was reported in nine studies (Berthelot [29,30,35,40,41,51,53,61]. The studies used Anxiety and Stress Scale-21 (DASS-21) [29,35,37,40,53], Perceived stress scale (PSS) [30], Post- traumatic stress disorder check list 5 (PCL-5) [41,51], the Pandemic-Related Pregnancy Stress scale (PREPS) [61], the Kessler Psychological Distress Scale (K10), [51], Hospital Anxiety, Depression and Stress scale (HADS) [42,46], Prenatal Distress Questionnaire (PDQ), Perceived Stress Scale (PSS) [35,36], Depression, Anxiety and Stress Scale-21 (DASS-21) [29]. One of those studies provided results on post-traumatic stress, comparing prevalence in pregnant and non-pregnant women (0.9% and 5.7%, respectively) [41]. Three studies showed distinctive prevalence rates of stress in pregnant women during the pandemic (11.1% in Singapore; 43.9% in Iran and 89.1% in China) [29,30,40]. The scale used by Heidi Preis (PREPS) indicated that pregnant women suffered stress related to prenatal preparation (27.2%) and the risk of prenatal infection (29.1%) [61]. Perceived stress was shown to increase with feelings of loneliness and fear of contagion and it appeared as a predictor in most anxious and depressive symptoms related to COVID-19 [35,53]. 

### 3.5. Mood and Psychopathological Symptoms

Social support was evaluated through two questionnaires: the social support effectiveness questionnaire (SSEQ) and the interpersonal support evaluation list (ISEL) (results obtained: 55.8, SD = 14.9 and 34.1, SD = 6.3 respectively) [38]. Insomnia measured though Insomnia severity index (ISI), was present in 2.6% of pregnant women and somatization problems were reported by 2.4% of pregnant women during the pandemic according to the “Somatization subscale of the symptom checklist 90 (SCL-90)” [41]. Results from another study showed that almost 20% of women interviewed suffered clinical insomnia (ISI > 15) (see cross-national study of factors associated with women’s perinatal mental health and wellbeing during the COVID-19 pandemic.) and another study revealed that insomnia was a predictor variable in most anxious and depressive symptoms related to COVID-19 [35]. In order to know the impact of physical activity on anxiety levels and depression of the pregnant women during the pandemic, the Godin Shephard Leisure-Time Exercise Questionnaire was used, the results obtained were 33.1 (SD = 21.2) [38]. Depersonalization problems were measured using two scales: Dissociative Experiences Scale (DES-II) (results obtained: dissociation/depersonalization: pre-pandemic: B = −0.17 (SD = 0.05), pandemic: 0.07 (SD = 0.03); Positive and Negative Affect Schedule (PANAS) (results obtained: negative affectivity: pre-pandemic: −0.64 (SD = 0.04), pandemic: 0.25 (SD = 0.03); low positive affectivity: pre-pandemic: −0.64 (SD = 0.04), pandemic: 0.25 (SD = 0.03) (Berthelot et al., 2020) and PANAS Positive: 28.71 (SD = 6.81); PANAS Negative: 22.61 (SD = 7.18) (Chaves et al., 2021). Finally, the results of Colli et al., 2021., reported that 11.2% was positive for obsessive-compulsive symptoms [53].

## 4. Discussion

The aim of this study was to describe the psychological impact of the COVID-19 pandemic on pregnant women and to identify the risk factors associated. To date, there are very few longitudinal studies comparing groups of pregnant women with other populations. The findings of this review indicate an increased prevalence of mental health symptomatology during pregnancy during the COVID-19 pandemic. The most prevalence psychological symptoms affecting pregnant women were depression, anxiety, stress and insomnia during the COVID-19 pandemic.

The main variables concerning expectant mothers were fear of using a public transport, staying in public places, the potential contagion of a relative, fear of infection, possible vertical transmission (from mother to the fetus), prematurity or miscarriage associated with the SARS-CoV-2 virus [21]. In turn, an adequate coping strategy is a key point for pregnant women. Variables such as having a good economic situation, an adequate level of education and sufficient social support [57,58] are protective factors against psychological problems. Social isolation and imposed quarantines tend to change the routines of pregnant women, leading to some of them not attending prenatal appointments [7,17]. This fact may increase the risk of adverse effects during pregnancy. Moreover, the exposure to biased information from the media and social networks has also caused pessimistic thoughts in expectant mothers and their relatives [23,58]. Nevertheless, one study indicated that pregnant women showed an advantage of facing mental problems caused by COVID-19, showing fewer depression, anxiety, insomnia, and PTSD symptoms than non-pregnant women [60]. This may be due, on the one hand, to the better previous situation in terms of mental health and socioeconomic status of women who decide to become pregnant, since there were significant differences between characteristics of participants such as age, marital status and occupation. 

The findings obtained from the 31 selected studies are in agreement with previous studies. A survey of pregnant women and postpartum women during the pandemic, showed that 40% of participants suffered from post-traumatic stress disorder and around 70% suffered from depression and/or anxiety following the onset of the new virus [62]. It is also in line with the results found in a systematic review that summarized the increase of mental disorders in pregnant women caused by this pandemic [63]. In contrast, previous studies have shown a prevalence of depression lower than 60% [43,45,55]. Nonetheless, these findings point out the importance of providing reliable information since one cause of these problems is the references that pregnant women receive [64]. A longitudinal study showed that the stress generated by COVID-19 is related to fear of contagion and its adverse effects [64]. Furthermore, previous studies pointed out that the stress generated during pregnancy, along with the risk of loneliness, increases the prevalence of depressive symptomatology [15]. Finally, another study points out that women concerns about being infected leads to use of disinfectant products in large quantities, which poses a danger of poisoning [64].

However, the COVID-19 pandemic has not been the first pandemic. The Spanish flu in the beginning of the XX Century and the H1N1 influenza pandemic in 2009 also affected pregnant women. Their levels of anxiety, stress and depression increased as a result of the situation they were exposed to. The measures imposed were similar, so that uncertainty about how the disease would affect the fetus and mother, social distance, the need to be alone at the time of delivery and reducing hospital stay were also responsible for mental problems in the peripartum period [65,66].

### 4.1. Limitations

The main limitation of this study is the small number of databases consulted and the lack of a meta-analytic perspective. Another issue that should be addressed in future investigations is the point of the pandemic at which the measurement was taken, since recent studies show a significant decrease in anxiety and depression symptoms after one year of its duration [67]. Moreover, the studies included in this review do not discriminate between the main stress factors according to Lancet’s COVID-19 Commission Mental Health Task Force recommendations: contracting the infection, close relation having the infection, safety of others in your care and stress of living in a pandemic [68]. Moreover, most of the included studies were cross-sectional self-report surveys, which could overestimate the real prevalence of this type of psychological disorders. In addition, geographical coverage of the articles selected was focused on high-income countries, which invites to infer that this outcome is a problem exclusive of those countries, which is far from the reality. Another limitation is the variation of the characteristics of the included studies: the sample sizes, the different types of the surveys implemented, the moment of the measurements and their reliability and validity. Finally, the selection bias hinders the ability to make a quantitative summary of the studies.

### 4.2. Strengths

The main strength is that we have conducted a scoping review following up the PRISMA extension for scoping reviews guidelines. Our search was exhaustive, collecting all kind of psychological symptoms. Finally, the point at which the measurement was taken must be taken into account, since the evolution of the pandemic and the restrictions adopted have been different in each region. For instance, the results of Aknin et al., 2021 showed a significant increase in anxiety and depression symptoms at the beginning of the pandemic and a significant decrease in these symptoms after one year of its duration. Nonetheless, the unexpectedly high rates of current mental health issues warrant an urgent call to action.

### 4.3. Implications for Practice and Future Research

These data could be helpful to guide future interventions or the adjustment of the pathways of care such as providing accurate information and encouraging pregnant women to engage in healthy behaviors during pregnancy [55,69]. Likewise, communication and reassurance about their routine prenatal care may be a priority to avoid increased levels of depression and anxiety. On the other hand, screening tests for psychological problems (e.g., EPDS, PHQ-9, GAD-7) and intimate partner violence should be implemented. It is essential to promote protective factors and positive coping strategies such as via internet-based mindfulness programs, daily routine, self-care, mindfulness, prenatal or postnatal groups. Another strategy is to provide online mental health resources and tele-psychotherapy to treat psychological problems in the perinatal period.

In order to enhance social support, health care providers may focus on the opportunity that families have, within subsystems and across the family, to buffer pregnant women against the risks of social disruption due to COVID-19 [70]. Clinicians may assess whether pregnant women have adequate social support and encourage them to have regular contact with relative and friends (via telephone, social networks, video calls or face-to-face when possible). Involving the partner during the perinatal period can also be of help. During the COVID-19 pandemic, certain psychological strategies have been reported to have some positive results at reducing anxiety, depression and stress among pregnant women [71]. More precisely, through an online cognitive-behavioral therapy during the COVID-19 pandemic, a group of psychologists managed to reduce a range of psychopathological symptoms in pregnant women. Reducing stress and psychopathological symptoms during pregnancy can also promote the health of the fetus [46]. This fact is associated with the Fetal Programming Hypothesis by which the environment the fetus is exposed to during pregnancy can shape his/her future health and disease after birth [72]. Another activity to promote is prolonged skin-to-skin contact with the infant and early and exclusive breastfeeding, whenever possible. 

The women exposed to previous pandemics (e.g., 1918 Spanish Flu) had a higher risk of having children that would prematurely die of a heart attack when adults [73]. A previous study based on more than 65 millions of women found that higher levels of stress during pregnancy is associated with obesity, infantile colic and autism spectrum disorder in the offspring [74]. 

Finally, this review helps to understand the mental health situation that pregnant women are facing as a result of the current pandemic.

## 5. Conclusions

The results of this scoping review appear to suggest that levels of anxiety, stress and depression in pregnant women have increased as a result of the COVID-19 pandemic. Nonetheless, these conclusions are drawn from observational studies conducted over a short period of time. Longitudinal studies with a more robust methodology are needed to confirm these results. The characteristics of the included studies also presented characteristics with wide methodological differences. Economic situation, education level and social support have a considerable impact on mental health in pregnant women. All pregnant women should have their psychosocial and mental health status assessed throughout pregnancy and postpartum. In turn, health-care providers may provide relevant and evidence-based information intended for both pregnant woman and their relatives, as well as promoting protective factors such as social support.

## Figures and Tables

**Figure 1 behavsci-11-00181-f001:**
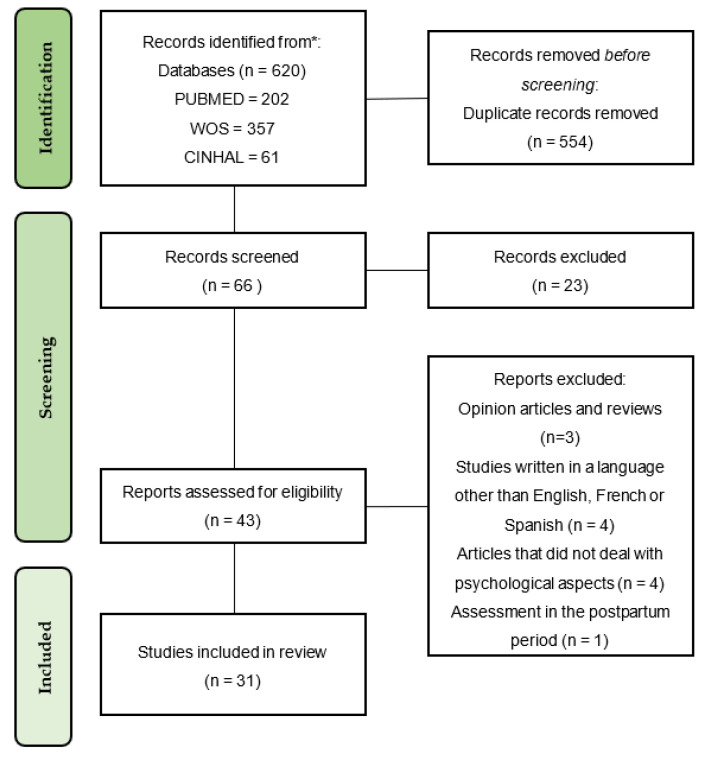
PRISMA 2020 Flow Diagram.

**Table 1 behavsci-11-00181-t001:** Inclusion and Exclusion Criteria.

Inclusion criteria	
Language	Studies written in Spanish, English or French
Design	Observational studies
Population	Pregnant women
Risk factor	Studies which considered the psychological issues of pregnant women during the COVID-19 pandemic
Year of publication	Articles published between 2020 and September 2021
Exclusion criteria	
Design	Literature reviews and editorials

**Table 2 behavsci-11-00181-t002:** Characteristics of included studies.

Author, Year	Country	Study Design	Assessment Tool	N	Age Mean (Years)	Gestational Age (Weeks)	Main Results	Conclusions	Quality of the Study *
Effati-Daryani et al., 2020 [29]	Iran	Cross sectional study	Questionnaire: DASS-21	205	29.3 (SD = 5.5)	NR	Pregnant women: 32.7% Depression 32.7% Anxiety 43.9% Stress (Prevalence)	Maintaining high partner satisfaction, possessing a high level of education (the couple) and a stable income level, decreases the levels of depression, anxiety and stress	8/10
Jiang et al., 2020 [30]	China	Cross sectional study	Questionnaire: PSS SAS EDS	1873	29 (SD = 4.10)	NR	Pregnant women: 45.9% Depression 18.1% Anxiety 89.1% Stress (Prevalence)	Using the informational tools provided by the hospital, decreases the risk of depression anxiety and stress.	6/10
Matsushima et al., 2020 [31]	Japan	Cross sectional	EPDS	1777	NR	Third trimester: 45.08% (SD= 0.50)	EPDS ≥ 13 17% (0.38), Depression: 1.82 (2.05), Anxiety: 3.68 (2.28)	A high percentage of pregnant women showed depressive symptoms	5/10
Shahid et al., 2020 [32]	Pakistan	Cross sectional	EPDS	552	NR	NR	39% stated that the pandemic had caused them depression and anxiety; 33% were found to have possible depression (EPDS > 10) and 6% scored EPDS = 30 (maximum depression)	Pregnancy is a determinant factor for negative perceptions of the COVID-19 pandemic	6/10
Sun et al., 2020 [33]	China	Cross sectional	EPDS	2883, Prenatal: 26.08%, Postnatal: 73.92%	25–29 aged: 41.42%	NR	The prevalence of maternal depression was increased from 30.99% to 42.98%	Prenatal depression in the beginning of the epidemic and postnatal depression in the end of the epidemic should be noticed	7/10
Wu et al., 2020 [34]	China	Cross sectional	EPDS	4124	30 (27-32)	NR	Overall, the prevalence of depressive symptoms was 26.0–29.6%.	The risk for mental illness among pregnant women have increased including thoughts of self-harm	9/10
Romero-Gonzalez et al., 2021 [35]	Spain	Cross sectional	SCL-90-R, PDQ, PSS, CD-RISC, AIS	131	32.95 (SD = 4.75)	27.20 (SD = 8.74)	Depressive symptomatology in confinement increases with loneliness, fear of contagion and perceived stress as well as anxiety, perceived stress and insomnia, increase with feelings of loneliness and fear of contagion	Perceived stress, pregnancy-specific stress, as well as insomnia are predictor variables in most anxious and depressive symptoms related to COVID-19.	3/10
Puertas-Gonzalez et al., 2021 [36]	Spain	Cross sectional	SCL-90-R, PSS, PDQ, AIS	Pregnant women during (PG) and prior (PPG) the pandemic (*n* = 100 respectively)	33.20 (SD = 4.71) and 33.04 (SD = 4.45)	26.47 (SD = 9.12) and 27.26 (SD = 8.70)	PG obtained higher scores than PPG with an average effect size in the depression dimension of the SCL-90, and with a small effect size in the phobic anxiety dimension of the SCL-90 and in the PSS	Women who were pregnant during the pandemic showed higher levels of depression and phobic anxiety than a group of pregnant women assessed before the COVID-19 pandemic	3/10
Yirmiya et al., 2021 [37]	Israel	Cross sectional	PHQ-2, PREPS, GAD-7	Pregnant (N = 1114), Non-Pregnant (N = 256)	31.88 (SD = 4.22), 35.71 (SD = 5.42)	NR	Pregnant women reported significantly fewer depressive symptoms than non-pregnant women. Non significant differences were observed between pregnant and non-pregnant women in levels of stress and anxiety	During the first wave of the COVID-19 pandemic, non-pregnant women had higher depressive symptoms than pregnant women.	4/10
Lebel et al., 2020 [38]	Canada	Cross sectional study	Questionnaire: EDS PROMIS SSEQ ISEL GSLTPAQ	1987	32.4 (SD = 4.2)	22.5 (SD = 8.4)	Pregnant women: 37% Depression 57% Anxiety (Prevalence)	Having high social support, staying physically active, is associated with lower levels of anxiety and depression in pregnant women during the pandemic.	5/10
Moyer et al., 2020 [39]	USA	Cross sectional study	Questionnaire: VAS PRaS	2740	32.7	NR	Pregnant women: PRaS score: pre-COVID-19: 20.6. Pregnant women: COVID-19: 23.9. (Prevalence)	The main concerns of pregnant women during the pandemic were: the fear of food shortages, the risk of contagion from work, the likelihood of not receiving care for the baby, etc.	6/10
QJ Ng et al., 2020 [40]	Singapur	Cross sectional	Questionnaire: DASS-21	324	31.8 (SD = 4.2)	23.4 (SD = 10)	Pregnant women: 18.2% Depression 35.8% Anxiery 11.1% Stress	Online platforms are the main source of information for pregnant women.	5/10
Zhou et al., 2020 [41]	China	Cross sectional	Questionnaire: PHQ-9, GAD-7, PCL-5, SCL-90, ISI	859 (544 pregnant;315 not pregnant)	Not pregnant: 35.4 (SD = 5.7). Pregnant: 31.1 (SD = 3.9)	NR	Pregnant women: 5.3% Depression. 6.8% Anxiety, 2.4% somatic symptoms, 2.6% insomnia, 0.9% post-traumatic stress	During the pandemic, pregnant women are less likely to suffer from anxiety and depression than women who are not.	5/10
Akgor et al., 2020 [42]	Turkey	Cohort	Questionnaire: HADS	297	27.64 (SD = 5.27)	27.04 (SD= 8.85)	HADS-A 7.94 (SD = 4.03), HADS-D 7.23 (SD = 3.84)	Anxiety and depresion were associated with concern about the inability to reach obstetrician and being in advanced age.	3/9
Farrell et al., 2020 [43]	Qatar	Cross sectional	GAD-7, PHQ-9	288	30,5 (SD = 5.3)	26.1 (SD= 14.3)	Prevalence of anxiety and depression was 34.4% and 39.2% respectively.	More than a third of women scored anxiety and depression ratings.	3/10
Khoury et al., 2021 [44]	Canada	Cross sectional	CWS, CES-D, ISI, and MSPSS	303	32.13 (SD = 4.22)	21.47 (SD = 8.92)	CWS 1.94 (SD = 0.97), CES-D: 11.50 (SD = 6.36), ISI score ≥ 15: 19.2% (clinical insomnia)	Pregnant women are experiencing high levels of anxiety and depressive symptoms during the COVID-19 pandemic	5/10
López-Morales et al., 2021 [45]	Argentina	Cohort	BAI, STAI-T	Pregnant women: 102, non-pregnant women: 102	32.56 (SD = 4.71)	32.56 (SD= 4.71)	Time 1 vs. Time 3: Pregnant women: BDI-II: 8.71 (6.08) vs. 15.42 (8.50), STAI-T: 22.66 (9.48) vs. 28.10 (9.60), Non pregnant women: BDI-II: 7.92 (4.53) vs. 10.83 (6.79), STAI-T: 21.51 (8.44) vs. 23.97 (9.27)	Pregnant women showed a more pronounced increase in depression, anxiety than the non-pregnant women.	7/9
Kahyaoglu -Sut and Kucukkaya, 2020 [46]	Turkey	Cross sectional	Questionnaire: HADS	403	28.2 (SD = 4.5)	27.9 (SD = 8.8)	Pregnant women: 64.5% Anxiety, 56.3% Depression	Being an active worker during the pandemic, maintaining a sports routine, information from a healthcare professional, are some of the beneficial factors related to anxiety and depression.	3/10
Ayaz et al., 2020 [47]	Turkey	Cross sectional	Questionnaire: BAI IDAS-II	63	30.35 (SD = 5.27)	32.7	Pregnant women: IDAS II: pre-COVID-19: Pre184.7 (SD = 49.67), COVID-19: 202.57 (SD = 52.90). BAI: ANYTHING: pre-COVID *n* = 10, COVID *n* = 6. SEVERE: Pre-COVID-19 *n* = 2, COVID-19 *n* = 8	Stress and anxiety levels in pregnant women have increased as a result of the COVID-19 pandemic	9/10
Taubman-Ben-Ari et al., 2020 [48]	Israel	Cross sectional	Questionnaire: HI-5	336	30.31 (SD = 4.7)	25.40 (SD = 9.6)	Causes of anxiety in pregnant women during the pandemic were the fear of being infected when attending gynecological appointments, using public transports or when walking in public places; fear of COVID-19 infection by a family member; worries about the fetus health; and the birth time.	Arab pregnant women showed higher levels of anxiety than Jewish women. In addition, suffering from poor health, being in the 3rd trimester of pregnancy and being primiparity were the factors that contributed to high anxiety levels.	4/10
Berthelot et al., 2020 [49]	Canada	Cohort	Questionnaire: K10 PCL-5 PANAS	1754	29.27 (SD = 4.23)	24.80 (SD = 9.40)	Pregnant women: Anxiety/depression: pre-COVID-19: -0.39 (SD = 0.04), COVID-19: 0.15 (SD = 0.03). Post-traumatic stress: pre-COVID-19: -0.12 (SD = 0.04). COVID-19: 0.06 (SD = 0.03).	Pregnant women surveyed during the pandemic suffer from higher levels of psychological events (anxiety and depression) than those surveyed before the pandemic.	5/9
Kassaw et al., 2020 [50]	Ethiopia	Cross sectional study	Interview GAD-7 OSLO-3	178	28 (SD = 5.6)	NR	Pregnant women: 32.2% Anxiety prevalence	Living in an urban environment, primiparity, secondary education and low social support, are risk factors for suffering from anxious symptoms.	10/10
Liu et al., 2020 [51]	China	Cross sectional study	Questionnaire: SAS	1947	NR	NR	Pregnant women: 17.2% Anxiety (Prevalence)	In Wuhan, more pregnant people suffered from anxiety than in Chongqing. Staying at home, having subjective symptoms increased anxious episodes.	6/10
Sinaci et al., 2020 [52]	Turkey	Cross sectional	Questionnaire: STAI-T, BAI	446	28.9 (SD = 5.7)	24.5 (SD = 7.7)	Pregnant women: Anxiety pre-pandemia: Total 39.34 (SD = 6.39), Anxiety during the pandemic: Total: 44.57 (SD = 9.55).	High-risk pregnant women suffer more anxiety than those without risk.	4/10
Colli et al., 2021 [53]	Italy	Cross sectional	PREPS, PSS, GAD-7, PHQ-2, OCD Screening	258	32.5 (SD = 5.12)	NR	32.6% reported clinically relevant anxiety levels and 11.2% was positive for OCD problems. Pandemic-related stress predicts the development of anxiety, depressive, and obsessive-compulsive symptoms.	The COVID-19 pandemic onset contributed to poor mental health, especially anxiety among Italian pregnant women	5/10
Preis et al., 2020 [54]	USA	Cross sectional	Questionnaire: PREPS	4451	30.8 (SD = 4.7)	27	Prenatal preparation stress: 27.2% Prenatal risk stress: 29.1%.	Having access to open spaces, not canceling prenatal appointments and leading a healthy life protect against stress	5/10
Chaves et al., 2021 [55]	Spain	Cross sectional	Questionnaire: EPDS and PANAS	N = 450 pregnancy, N = 274 postpartum	33.36 (SD = 4.12)	NR	Total EPDS: 12 (SD = 5.19); EPDS-Anxiety: 7.36 (SD = 2.47); PANAS Positive: 28.71 (SD = 6.81); PANAS Negative: 22.61 (SD = 7.18).	58% and 51% of women reported depressive and anxiety symptoms respectively.	2/10
Davenport et al., 2020 [56]	Canada	Cross sectional	EPDS, STAI-State	N = 520 pregnancy, N = 380 postpartum	33 (SD = 8)	NR	Pre-pandemia vs. Pandemia: EPDS: 7.5 ± 4.9 vs. 11.2 ± 6.3; STAI = 34.5 ± 11.4 vs. 48.1 ± 13.6	40.7% and 72% of women reported an indicative of depression and moderate to high anxiety respectively.	3/10
Mappa et al., 2020 [57]	Italy	Cross sectional	STAI-T	200	33 (IQR 30–36)	18 (IQR 15–23)	STAI-T: 37 (IQR 20–43) STAI-T ≥ 40: 38.2% CI 31.3–45.5)	COVID 19 induced a significant increase in maternal anxiety	3/10
Saccone et al., 2020 [58]	Italy	Cross sectional	STAI, IES-R	100			IES-R: 36.9 (10.1), STAI 45.2 (14.6)	COVID-19 outbreak had a moderate to severe psychological impact on pregnant women.	3/10

BAI: Beck anxiety inventory, IDAS-II: Inventory of Depression and Anxiety Symptoms II, MHI-5: Mental Health Inventory- Short Form, K10: Kessler Psychological Distress Scale, PCL-5: Post traumatic stress disorder check list 5, PANAS: Positive and Negative Affect Schedule, GAS-7: General Anxiety Disorder, OSLO-3: Psychometric properties of the 3-item Oslo social support scale among clinical students, DASS-21: Depression, Anxiety and Stress Scale-21, SAS: Self Rating Anxiety Scale, EDS: Edinburg depression scale, PROMIS: Patient-Reported Outcomes Measurement Information System, SSEQ: social support effectiveness questionnaire, ISEL: Interpersonal support evaluation list, GSLTPAQ: Godin Shephard Leisure-Time Exercise Questionnaire, VAS: Visual analog scale, PRaS: A modified pregnancy-related anxiety scale, PREPS: Pandemic-related Pregnancy Stress Scale, STAI-T: Spielberg State trait anxiety inventory trait subscale, HADS: Hospital Anxiety, Depression and Stress scale, PHQ-9: Patient Health questionnaire, GAD-7: General Anxiety Disorder, SCL-90-(R): Somatization subscale of the symptom checklist 90 (Revised), ISI: Insomnia severity index; SWLS: Satisfaction With Life Scale; EPDS: Edinburgh Postnatal Depression Survey, STAI-State: State-Trait Anxiety Inventory, CWS: Cambridge Worry Scale, CES-D: Centre for Epidemiologic Studies Depression Scale, MSPSS: Multidimensional Scale of Perceived Social Support, IES-R: Impact of Event Scale-Revised, PDQ: Prenatal Distress Questionnaire, PSS: Perceived Stress Scale, CD-RISC: Connor-Davidson Resilience Scale, AIS: Athens Insomnia Scale NR: Not reported. * Quality of each study was assess using the Newcastle-Ottawa Quality Assessment Scale (NOQAS).

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
