# Peer review of "Psychological Impact of the COVID-19 Pandemic on Pregnant Women: A Scoping Review"

_behavsci, 2021, doi:10.3390/bs11120181_

Round 1

Reviewer 1 Report

As long as I am not an expert in mental symptoms manifested by pregnant women, my report will be only focusing on methodological issues and approaches.

  • A stronger rationale is needed to explain the selection of the methodology beyond the explanation that the scoping review is systematic and rigorous. The same could be said about systematic reviews.
  • Authors should made explicit the research question/s that guided their review. Right now there is not a clear research question.
  • More information is need about how the quality of the studies was evaluated. Authors should offer a more detailed explanation of the NOQAS and an example of how studies were evaluated to understand the punctuation included in Table 1.
  • Inclusion and exclusion criteria should be further elaborated. For example, a table with all the inclusion and exclusion criteria could be a good addition to improve readiness of the methodology.
  • Authors should further detail reason for exclusion of the 12 article in the screening phase.
  • Author should indicate the language of the 31 articles selected for final analyses.
  • Given that all the studies analysed are cross-sectional, authors should temper their conclusion.

Reviewer 2 Report

Thank you for your submission and the opportunity to review. My concerns are-

  1. Only list one time period for search- start month and ending month/year- no need to say repeated search was conducted- November to January to September, list only November 2020 to September 2021.
  2. Limit your assumptions- you mention heterogeneity but you dont analyze studies for heterogeneity.
  3. Arrange studies in the table by when data was collected in chronological order- first to last.
  4. Remove the married/marriage column in the table- it has missing information and does not add much to the paper.
  5. For column 2 in table 1- only list country as you have this information for all studies- remove city information as this is not relevant and there are missing cities for some studies.
  6. List all limitations of the studies- e.g. variation in sample sizes, survey types, measures, reliability and validity issues with studies included, selection bias, lack of an ability to do a summary of quantitative studies like we do in meta-analysis.
  7. Check language, unexplained abbreviations, grammar, etc.
  8. Build a better section for implications for practice, future research, prevention, clinical care.
  9. Expand on conclusions with what you did, how, what you found and only say what you can definitely prove.

Reviewer 3 Report

The aim of this study was to describe the impact of the COVID-19 pandemic on pregnant women and to identify the risk factors that cause the most common psychiatric disorders in pregnant women. 

The presented work is valuable because it addresses a very important social problem and, what is important, it contains the results of a search of databases with articles not only in English. Thanks to the use of PRISMA protocol, the paper is readable and meets all requirements for review articles.
 The results of this review indicate an increased prevalence of mental health symptomatology during pregnancy as a result of covid-19 pandemic. In particular, this review found that in studies during covid-19 pandemic there was an increase in levels of anxiety-depressive symptoms, stress and insomnia. The authors relate their findings to the situation created by the H1N1 or Spanish flu epidemics. In my opinion, the main limitation of the study is the lack of meta-analytical data, but the authors are aware of this. Despite this limitation, the paper is a valuable contribution to the topic and should be published. 

Round 2

Reviewer 1 Report

The manuscript has been significantly improved. However, the research question posed is not a research question. Please, revise in order to express the objective inserted at the end of the introduction in a question format and indicate was was expected to find through the review planned.

Author Response

The authors would like to express their sincere thanks to the reviewer whose suggestions have improved our manuscript.

Regarding the last comment made: "Please revise in order to express the objective inserted at the end of the introduction in a question format and indicate was expected to find through the planned review". 

The authors of the manuscript agree completely. It is required to specify the research question and the objective of the present study. The following modification has been made at the end of the introduction: 

"The objective of this study was to assess the psychological impact of the COVID-19 pandemic on pregnant women and to estimate the prevalence of mental symptoms in this population".

Thank you very much for taking the time to review this work.

Best regards, 

Blanca Riquelme

Reviewer 2 Report

Thank you for the revisions.

Author Response

The authors would like to express their sincere thanks to the reviewer whose suggestions improved our manuscript.

Thank you very much for taking the time to review this work.

Round 3

Reviewer 1 Report

I endorse publication

This manuscript is a resubmission of an earlier submission. The following is a list of the peer review reports and author responses from that submission.

Round 1

Reviewer 1 Report

Thank you to the editor and the authors for the opportunity to review this manuscript. The authors examine the impact of the COVID-19 pandemic on pregnant women's mental health. However I see essential shortcomings of the study methods and procedure that have severe impact on the data interpretation. I summarize the main problem in the following and provide more details in my major comments below.

The biggest objection to this review relates to an error in the article screening procedure. The search results omitted at least 12 articles on this topic:

  1. Akgor, U.; Fadıloglu, E.; Soyak, B.; Unal, C.; Cagan, M.; Temiz, B.E.; Ak, S.; Gultekin, M.; Ozyuncu, O. Anxiety, depression and concerns of pregnant women during the COVID-19 pandemic. Gynecol. Obstet. 2021, 1–6.
  2. Chaves, C.; Marchena, C.; Palacios, B.; Salgado, A.; Duque, A. Effects of the COVID-19 pandemic on perinatal mental health in Spain: Positive and negative outcomes. Women Birth 2021.
  3. Davenport, M.H.; Meyer, S.; Meah, V.L.; Strynadka, M.C.; Khurana, R. Moms are not ok: COVID-19 and maternal mental health. Glob. Women’s Health 2020, 1, 1.
  4. Farrell, T.; Reagu, S.; Mohan, S.; Elmidany, R.; Qaddoura, F.; Ahmed, E.E.; Corbett, G.; Lindow, S.; Abuyaqoub, S.M.; Alabdulla, M.A. The impact of the COVID-19 pandemic on the perinatal mental health of women. Perinat. Med. 2020, 48, 971–976.
  5. Khoury, J.E.; Atkinson, L.; Bennett, T.; Jack, S.M.; Gonzalez, A. COVID-19 and mental health during pregnancy: The importance of cognitive appraisal and social support. Affect. Disord. 2021, 282, 1161–1169.
  6. López-Morales, H.; Del Valle, M.V.; Canet-Juric, L.; Andrés, M.L.; Galli, J.I.; Poó, F.; Urquijo, S. Mental health of pregnant women during the COVID-19 pandemic: A longitudinal study. Psychiatry Res. 2020, 295
  7. Mappa, I.; Distefano, F.A.; Rizzo, G. Effects of coronavirus 19 pandemic on maternal anxiety during pregnancy: A prospectic observational study. Perinat. Med. 2020, 48, 545–550.
  8. Matsushima, M.; Horiguchi, H. The COVID-19 pandemic and mental well-being of pregnant women in Japan: Need for economic and social policy interventions. Disaster Med. Public Health Prep. 2020, 1–6.
  9. Saccone, G.; Florio, A.; Aiello, F.; Venturella, R.; De Angelis, M.C.; Locci, M.; Bifulco, G.; Zullo, F.; Sardo, A.D.S. Psychological impact of coronavirus disease 2019 in pregnant women. J. Obstet. Gynecol. 2020, 223, 293–295.
  10. Shahid, A.; Javed, A.; Rehman, S.; Tariq, R.; Ikram, M.; Suhail, M. Evaluation of psychological impact, depression, and anxiety among pregnant women during the COVID-19 pandemic in Lahore, Pakistan. J. Gynecol. Obstet. 2020, 151, 462–465.
  11. Sun, G.; Wang, F.; Cheng, Y. Perinatal depression during the COVID-19 epidemic in Wuhan, China. China 2020.
  12. Wu, Y.; Zhang, C.; Liu, H.; Duan, C.; Li, C.; Fan, J.; Li, H.; Chen, L.; Xu, H.; Li, X.; et al. Perinatal depressive and anxiety symptoms of pregnant women during the coronavirus disease 2019 outbreak in China. J. Obstet. Gynecol. 2020, 223, 240-e1

As this is almost half of the articles included in the review, it is difficult to make it mandatory. Moreover, many of these studies show significantly lower prevalence of depression than 60%.

Furthermore, as there are already reviews on this subject in the literature (Brooks, at al., 2020; Ahmad & Vismara, 2021), the authors did not indicate at all why their review adds something more to our knowledge. I even have the impression that this review is misleading by over-interpreting many of the results (more in major comments). Unfortunately, the article in this shape is not suitable for publication.

Major comments:

- the headings in the results section must be changed. DEPRESSION DISORDER section does not described the depression disorder. The results of the studies indicate only the symptoms of depression and not the diagnosis of a depressive disorder. All reported studies are based on self-report methods. ANXIETY DISORDER section also does not describe any anxiety disorders. Moreover, the authors themselves describe the worries of pregnant women, not the increase in anxiety disorders. STRESS DISORDER heading is misleading. First, there is no such disorder in the ICD-10 or DSM-5 classification. Second, do the authors really describe any disorders in this section?

- line 168: "one of these studies reported increased post-traumatic stress levels in pregnant women...". This study only shows a higher score on the anxiety symptom scale, not the level of PTSD. In this study again, there were no diagnosis of PTSD symptoms, including reliving the trauma.

- line 171: Why is there no reference in the discussion to the results of the study that pregnant women have a much lower level on the PTSD scale than non-pregnant women?

- line 231: "A survey of pregnant women and postpartum women during the pandemic, conducted by Koenen et al., 2020, showed that 40% of participants suffered from post-traumatic stress disorder and around 70% suffered from depression and/or anxiety following the onset of the new virus" - please provide the citation to this research.

- line 268: "levels of anxiety, stress and depression in pregnant women have skyrocketed as a result of the COVID-19 pandemic". Can such a conclusion be made at all on the basis of a narrative review?

- line 255: Justification that despite the limitations rapid review is needed, due to the urgent pandemic situation is insufficient. Firstly, there are already such reviews, and secondly, this review does not take into account the results that show a different picture of the pandemic impact on the mental health. What we currently need is a faithful picture and careful analysis, not quick overviews that can be misleading.

- Table 1: In the authors column there is the first author's name and surname - please correct this to the citied version, so that the name is at the beginning, and second, that there is an indication of the other authors (i.e. adding "et al.")

- Figure 1: change cause to "due to"

In summary, new reviews should evaluate the impact of symptom measurement tools on the research results; methods of measurement (on-line survey or face-to-face measurement); distinguishing the participants stress factors (4 layer of potential stress during the COVID-19 pandemic) according to the Lancet's COVID-19 Commission Mental Health Task Force recommendations; the point at which the measurement was taken (research shows a significant increase in anxiety and depression symptoms at the beginning of the pandemic and a significant decrease in these symptoms after one year of its duration) (Aknin et al. 2021; doi: 10.31234/osf.io/zw93g). I hope that this tips will be valuable for the authors when creating a new review, which is definitely needed in this topic.

Reviewer 2 Report

This study reviews the impact of Covid-19 on pregnant women. This is an important topic in public health.  But some questions still need to be answered:

  1. The research design of this paper is simple. It is recommended to add a literature comparison of the mental health of pregnant women before and after the Covid-19 pandemic.
  2. It is suggested to raise a  framework to descript the relationship between the factors and psychopathological health of pregnant women.
  3. The number of documents used for analysis is small, and the scope of document retrieval should be broadened. For example, add mental health and psychology as the search term.
  4. Future study designs basd on this review need to be added.